# PARP Theranostic Auger Emitters Are Cytotoxic in BRCA Mutant Ovarian Cancer and Viable Tumors from Ovarian Cancer Patients Enable Ex-Vivo Screening of Tumor Response

**DOI:** 10.3390/molecules25246029

**Published:** 2020-12-19

**Authors:** Aladdin Riad, Sarah B. Gitto, Hwan Lee, Harrison D. Winters, Paul M. Martorano, Chia-Ju Hsieh, Kuiying Xu, Dalia K. Omran, Daniel J. Powell, Robert H. Mach, Mehran Makvandi

**Affiliations:** 1Division of Nuclear Medicine and Clinical Molecular Imaging, Department of Radiology, Perelman School of Medicine, University of Pennsylvania, Philadelphia, PA 19104, USA; Aladdin.Riad@pennmedicine.upenn.edu (A.R.); Hwan.Lee@pennmedicine.upenn.edu (H.L.); harrisonwinters@gmail.com (H.D.W.); Paul.Martorano@pennmedicine.upenn.edu (P.M.M.); chiahs@pennmedicine.upenn.edu (C.-J.H.); kuxu@pennmedicine.upenn.edu (K.X.); rmach@pennmedicine.upenn.edu (R.H.M.); 2Ovarian Cancer Research Center, Division of Gynecology Oncology, Department of Obstetrics and Gynecology, Perelman School of Medicine, University of Pennsylvania, Philadelphia, PA 19104, USA; Sarah.Gitto@pennmedicine.upenn.edu (S.B.G.); domran@pennmedicine.upenn.edu (D.K.O.); poda@pennmedicine.upenn.edu (D.J.P.J.); 3Center for Cellular Immunotherapies, University of Pennsylvania, Philadelphia, PA 19104, USA; 4Abramson Cancer Center, Department of Pathology and Laboratory Medicine, Perelman School of Medicine, University of Pennsylvania, Philadelphia, PA 19104, USA

**Keywords:** Auger therapy, radiopharmaceutical therapy, theranostic, PARP-1, ovarian cancer, BRCA1, HRD

## Abstract

Theranostics are emerging as a pillar of cancer therapy that enable the use of single molecule constructs for diagnostic and therapeutic application. As poly adenosine diphosphate (ADP)-ribose polymerase 1 (PARP-1) is overexpressed in various cancer types, and is localized to the nucleus, PARP-1 can be safely targeted with Auger emitters to induce DNA damage in tumors. Here, we investigated a radioiodinated PARP inhibitor, [^125^I]KX1, and show drug target specific DNA damage and subsequent killing of *BRCA1* and non-*BRCA* mutant ovarian cancer cells at sub-pharmacological concentrations several orders of magnitude lower than traditional PARP inhibitors. Furthermore, we demonstrated that viable tumor tissue from ovarian cancer patients can be used to screen tumor radiosensitivity ex-vivo, enabling the direct assessment of therapeutic efficacy. Finally, we showed tumors can be imaged by single-photon computed tomography (SPECT) with PARP theranostic, [^123^I]KX1, in a human ovarian cancer xenograft mouse model. These data support the utility of PARP-1 targeted radiopharmaceutical therapy as a theranostic option for PARP-1 overexpressing ovarian cancers.

## 1. Introduction

Theranostics provide a cutting-edge approach to cancer therapy by utilizing a single agent that can diagnose, monitor, and treat a patient’s disease. Auger and alpha emitting radionuclides have high-linear energy transfer (high-LET) properties capable of inducing DNA damage more efficiently than low-LET gamma or beta radiation [1]. Although alpha-emitters are known to cause high levels of DNA damage, very few radionuclides have imageable photons in their decay chain [2]. For these reasons, theranostic Auger emitters offer a unique portfolio of radionuclides that have imageable properties through positron or single-photon emission tomography and therapeutic properties from Auger electrons [3].

Several small molecules have been investigated to deliver therapeutic Auger radionuclides, including iodine-123, iodine-125, and bromine-77. These agents target various proteins or processes, including direct DNA incorporation ([^125^I]IUDR, [^77^Br]BrUDR), prostate-specific antigens ([^125^I]DCIBzL), and estrogen receptors ([^123^I]iodo-1,1-bis(4-hydroxyphenyl)-2-phenylethylene) [4,5,6,7]. These studies have shown the initial proof of concept for utilizing Auger emitters as theranostic radionuclides; however, with the exception of [^125^I]IUDR and [^77^Br]BrUDR, the molecular targets are not located within the nucleus or in close proximity to DNA. Furthermore, deoxyuridine analogues have shown in-vivo deiodonation/debromination, which suggests instability of the construct. Recently, poly (ADP-ribose) polymerase (PARP) targeted beta and Auger emitting radiopharmaceuticals have been reported for the treatment of glioblastoma and showed anti-tumor effects both in-vitro and in-vivo [8,9]. Furthermore, other bromine-77 PARP inhibitors have been reported, although pre-clinical evaluation is ongoing [10,11].

The path-length of Auger electrons is 10–500 nm [3], requiring delivery of Augur emitters directly or in close proximity to DNA to induce damage and cancer cell death. Poly (ADP-ribose) polymerase 1 (PARP-1) is an ADP-ribosylation enzyme that is essential for DNA damage response and is overexpressed in most tumor types [12]. PARP-1 subcellular localization on the chromatin provides an ideal target for tumor cell specific delivery of Auger emitters to DNA [13].

In the present study, we investigated PARP-1 as a unique molecular target that is highly expressed in cancer nuclei for the delivery of theranostic Auger emitting radionuclides. Recent work has demonstrated excellent visualization of nuclear localization of the PARP inhibitor rucaparib through fluorescent imaging, which is a close analogue to KX1 reported in this work, supporting a strong rationale for this approach [14]. Herein, we report in-vitro mechanistic studies exploring the specificity of DNA damage induced by a previously reported radioiodinated PARP inhibitor, KX1, labeled with Auger emitting radionuclides iodine-125 and iodine-123. Studies were performed using viable patient tumor slice cultures as a means for screening therapeutic efficacy ex-vivo. Furthermore, we performed in-vivo proof of concept studies to highlight the theranostic potential of PARP inhibitors radiolabeled with theranostic Auger emitters.

## 2. Results

### 2.1. PARP-1 Expression in Breast and Ovarian Cancer

RNA sequencing data from The Cancer Genome Atlas showed that PARP-1 is overexpressed in ovarian and breast carcinoma compared to normal tissue. Furthermore, 1132 breast (grey dotted box) and 420 ovarian cancer (black dotted box) patients’ tumors had significantly higher expression of PARP-1 in both cancers compared to all normal tissues except bone marrow and spleen (one-way ANOVA; *p*-value < 0.05; Figure 1). This demonstrates that PARP-1 may serve as a suitable drug target for Auger therapy in breast and ovarian cancer patients that have high PARP-1 expressing tumors.

### 2.2. Radiochemistry

[^123/125^I]KX1, an Auger emitting analogue of rucaparib, was synthesized through electrophilic aromatic destannylation [15]. The final product was isolated with radiochemical recovery yields of 60–70% and had a chemical and radiochemical purity >90%. Final products were diluted with either cell culture media or saline with <1% ethanol in the final solution for respective applications in this study. [^123^I]KX1 was evaluated as a theranostic imaging agent and [^125^I]KX1 was used to study the mechanism of action of Auger emitters in-vitro.

### 2.3. [^125^I]KX1 Targeted Cell Killing is Dependent on PARP-1 Binding

We first assessed the cell viability of previously reported OVCAR8 isogenic polyclonal cell lines that include OVCAR8 wild type (wt), OVCAR8 cas9, and three PARP1 knock-out cell lines (KO) [16] after treatment with [^125^I]KX1 for 1 h followed by 7 days of culture, and found that OVCAR8 cells were more sensitive to [^125^I]KX1 than PARP1 KO cells (Appendix A). OVCAR8 wt and OVCAR8 cas9 cells were more sensitive than OVCAR8 PARP1 KO cells (effective concentration for 50% reduction in cell viability (EC_50_) 13.6 ± 0.629 for wt, 12 ± 0.7 for cas9 vs. 46.6 ± 4 for KO g1, 36 ± 3.2 KO g2, 57.8 ± 4 KO g3 kBq/mL) (Appendix A). To verify that the cytotoxic effects observed were selective, [^125^I]*meta*-iodobenzylguanidine (MIBG) was used as a nonspecific iodine-125 control, as it is not directed towards PARP-1 and would not be internalized by ovarian cancer cells due to the lack of norepinephrine transporter expression. Results show that [^125^I]MIBG alone or in combination with non-radiolabeled KX1 were >10 times less cytotoxic than [^125^I]KX1 values in both wild type and PARP1 KO cells (Appendix A).

To further characterize [^125^I]KX1 binding to PARP-1, saturation radioligand binding studies were performed on OVCAR8 wt and PARP1 KO cells. We determined a K_d_ value of 7.7 nM with a B_max_ value of 1.2 × 10^6^ sites/cell for OVCAR8 wild type cells, and a K_d_ value of 7.3 nM with a B_max_ value of 3.9 × 10^5^ sites/cell for OVCAR8 PARP1 KO cells (Appendix A). These data showed that [^125^I]KX1 binds to PARP-1 in a specific manner, and further confirmed that the PARP1 KO cells have a reduction in PARP-1 expression, which corresponds to decreased sensitivity. Competition radioligand binding assays were performed to show that veliparib, the least toxic PARP inhibitor under clinical development, was able to compete with [^125^I]KX1 for binding with PARP-1. We determined that veliparib has a K_i_ value of 9.4 nM in the wt cells and a K_i_ value of 10 nM in the PARP1 KO cells (Appendix A). This supported that veliparib was a suitable control for our treatment studies as it blocks [^125^I]KX1 binding to PARP-1, preventing localization to DNA at concentrations that are non-toxic to cells.

### 2.4. [^125/123^I]KX1 Induces DNA Damage

To evaluate the mechanism of [^125^I]KX1 cytotoxicity, we performed immunofluorescent cell microscopy studies to analyze DNA damage marker γH2AX foci formation in OVCAR8 and OVCAR8 PARP1 KO cells. Cells were treated with 0.925, 1.85 or 3.7 MBq/mL [^125^I]KX1 for 1 h prior to immunofluorescent staining. [^125^I]KX1 caused a dose-dependent increase in γH2AX foci that was PARP-1 specific after treatment (Figure 2A) and could be pharmacologically blocked with veliparib (Figure 2A). Statistically significant increases in DNA damage induced by [^125^I]KX1 treatment were found in all doses evaluated in OVCAR8 wild type cells (one-way ANOVA; *p*-value < 0.0001) where only a significant effect was observed at the high dose in PARP1 KO cells (one-way ANOVA; *p*-value < 0.001) (Figure 2B). A similar effect was observed in cells treated with [^125^I]KX1 or [^123^I]KX1 for 2 h (Appendix A). Veliparib blocking studies were performed to show that the induction of DNA damage is specific to [^125^I]KX1 binding to PARP-1. Cells co-treated with a non-toxic dose of veliparib and increasing doses of [^125^I]KX1 showed blocking PARP-1 with veliparib inhibited DNA damage in the presence of [^125^I]KX1 (Figure 2B). Taken together, these results indicate that the DNA damage induced by [^125^I]KX1 is dependent on availability of PARP-1 for drug-target engagement.

Next, we showed the difference in DNA damage induced with continuous exposure to [^125^I]KX1 for 24 h vs. a 2 h exposure followed by a washout with a 24 h treatment free period that allowed for DNA repair. γH2AX foci formation was greater in both treatment groups for OVCAR8 cells when compared to the OVCAR8 PARP1 KO cells although all groups showed statistically significant increases from control (one-way ANOVA; *p*-value < 0.05) (Appendix A). Furthermore, 24 h treatment of [^123^I]KX1 or 10µM olaparib in OVCAR8 and OVCAR8 PARP1 KO cells showed similar results (Appendix A). We found that both [^123^I]KX1 and olaparib treatments resulted in reduced γH2AX foci in OVCAR8 PARP1 KO compared to OVCAR8 cells, although all groups showed statistically significant differences from control (one-way ANOVA; *p*-value < 0.05). This further highlighted the specificity of [^123^I]KX1 induced DNA damage to cells expressing PARP-1 and showed that both targeted agents required PARP-1. In addition we found γH2AX foci formation was increased for cells treated with [^123^I]KX1 compared to olaparib treatment. This illustrates that [^123/125^I]KX1 induces DNA damage in a radiation dose-dependent manner after just 1 h of treatment.

### 2.5. [^123/125^I]KX1 Does Not Inhibit the Enzymatic Activity of PARP-1

Since the chemical structure of [^123/125^I]KX1 is a derivative from a PARP inhibitor, it was important to assess whether the measured DNA damage was specific to the targeting of PARP-1 with Auger emitting radionuclides and not due to the biochemical inhibition of PARP-1. OVCAR8 cells treated with 4 MBq/mL, a quantity known to induce DNA damage by accumulation of yH2AX, did not inhibit the biochemical production of poly(ADP-ribose) (PAR), confirming that PARP-1 enzymatic activity was not inhibited by [^123^I]KX1 and is in agreement with the radiotracer principle. OVCAR8 cells treated with 10 µM olaparib were used as a positive control for the biochemical inhibition of PARP-1. We analyzed PAR using semi-quantitative immunofluorescence as a surrogate marker of PARP-1 enzymatic activity to confirm that [^123/125^I]KX1 did not cause PARP-1 catalytic inhibition as seen with olaparib. Interestingly, [^125^I]KX1 treatment for 24 h, or for 2 h followed by a 24 h washout period, did not inhibit PAR but increased PAR from control cells at 24 h (Appendix A) (one-way ANOVA; *p*-value < 0.05) and is consistent with enzymatic activation of PARP-1 as part of DNA damage response signaling [17]. This was in contrast to OVCAR8 cells treated with 10 µM olaparib, which showed a reduction in PAR (Appendix A) (one-way ANOVA; *p*-value < 0.01). These data corroborate our findings in fluorescent microscopy experiments suggesting that DNA damage is dependent on PARP-1 expression and independent from catalytic inhibition of PARP-1. Furthermore, this provides evidence that Auger emitting radionuclides delivered to the nucleus of the cells via PARP-1 targeting induced DNA damage at sub-pharmacological levels, which is consistent with the radiotracer principle.

### 2.6. In-Vitro Dosimetry and Relative Biological Effectiveness of [^125^I]KX1

When evaluating [^125^I]KX1 compared to the non-radioactive PARP inhibitor rucaparib on a molar scale, [^125^I]KX1 is 1000 times more potent. Since tumors with homologous recombination DNA repair deficiencies (HRD), such as a BRCA1 mutation, have increased sensitivity to PARP inhibitors, we assessed whether such mutations also increase sensitivity to [^125^I]KX1. Dose–response curves comparing two isogenic ovarian cancer cell lines, UWB1.289-BRCA1 mutated and UWB1.289-BRCA1 restored, show a leftward shift in the dose–response curve for UWB1.289 (radiation dose for 50% survival (D_50_) 2.31 ± 0.21 Gy) compared to UWB1.289-BRCA1 restored (3.45 ± 0.36 Gy) (Figure 3A). Similar results were obtained for OVCAR8 (BRCA1 methylated; D_50_ 1.41 ± 0.04) and SNU-251 (BRCA1 mutant; D_50_ 1.68 ± 0.10 Gy) compared to SKOV3 (BRCA1 wild type; D_50_ 3.59 ± 0.32 Gy) cell lines (Figure 3A). Analysis of OVCAR8 wild type (D_50_ 1.41 ± 0.04 Gy) and OVCAR8 PARP1 KO (D_50_ 1.26 ± 0.07 Gy) cells was performed concurrently as a comparison and when the radiation dose was normalized by incorporating PARP-1 expression, only small differences in radiosensitivity were observed (Figure 3A). These data provide further evidence that [^125^I]KX1 cellular lethality is PARP-1 specific and is more effective in HRD cancer cells.

In-vitro dosimetric analysis revealed that the relative biological effectiveness (RBE) of the Auger-emitting [^125^I]KX1 in comparison to low-LET beta emitting [^131^I]KX1 was lowest in BRCA1 mutant ovarian cancers SNU-251 and UWB1.289 (Figure 3B). Functional restoration of BRCA1 in the UWB1.289 isogenic cell line showed an increase in RBE from 0.7 to 1.78. Interestingly, the resistance enhancement (BRCA1 restored D_50_/BRCA1 mutant D_50_) of the BRCA1 mutation in UWB1.289 isogenic cell lines was only 1.49 for [^125^I]KX1 but 3.79 for [^131^I]KX1. Cell lines with the highest RBE included OVCAR8 isogenic cell lines and SKOV3. The RBE of [^125^I]KX1 was less than one for UWB1.289, which was more sensitive to [^131^I]KX1 (D_50_ 1.62 ± 0.08 Gy). Despite the differences observed, the average RBE for all cell lines was greater than three, which is consistent with high-LET properties of Auger radiation. In comparison to relative sensitivity of clinically used PARP inhibitors, the RBE of [^125^I]KX1 increased as sensitivity to PARP inhibitors decreased. HRD cell lines were more sensitive to [^125^I]KX1 than non-HRD cell lines (*t*-test, *p* < 0.05); however, statistically significant differences were not found for [^131^I]KX1 although the data were trending similarly (Figure 3C).

### 2.7. [^125^I]KX1 Effects on Patient Tumors

To determine if [^125^I]KX1 induced apoptosis or DNA damage in human patient tumors, viable tumor slices obtained from a clinical biopsy were cultured with increasing doses of [^125^I]KX1, and with [^125^I]KX1 in the presence of veliparib to block [^125^I]KX1 induced DNA damage. Immunohistochemical analysis of PARP and cleaved-PARP, a marker for apoptosis, showed [^125^I]KX1 treatment increased the expression of tumor cell specific cleaved-PARP in a dose-dependent manner (Figure 4A,B). At the 3.7 MBq/mL dosage, a significant increase in cleaved-PARP was observed (one-way ANOVA; *p* = 0.0394). Consistent with what we observed in our cell culture model system, veliparib inhibited the effects of [^125^I]KX1. Additionally, tumor slices treated with [^125^I]KX1 had increased expression of yH2AX as determined by confocal microscopy (Figure 4C).

### 2.8. SPECT/CT Imaging, Ex-Vivo Autoradiography and Tissue Histology

[^123^I]KX1 microSPECT/CT imaging showed tumor specific uptake (Figure 5A). Following microSPECT/CT imaging, animals had tumors and muscle controls immediately resected, snap frozen and sectioned at a thickness of 30 µm for ex-vivo autoradiography (Figure 5B). Tumor to muscle ratios were calculated using three independent methods including SPECT/CT (1.29 ± 0.05, *n* = 3), gamma counting (2.1 ± 0.15, *n* = 3), and autoradiography (3.0 ± 0.21, *n* = 3) (Figure 5C). These results are consistent with previously reported biodistribution studies performed in breast cancer models [18]. The higher tumor to muscle ratio most likely comes from the increased sensitivity of autoradiography vs. microSPECT imaging. Finally, adjacent portions of the tumors resected for ex-vivo autoradiography were analyzed histologically for DNA damage using RPA staining. An increase in DNA damage in mice treated with 29.6 MBq of [^123^I]KX1 compared to untreated was observed (Figure 5D). Control muscle samples that did not include OVCAR8 tumor cells showed no significant increase in RPA immunofluorescence in mice treated with 29.6 MBq of [^123^I]KX1.

## 3. Discussion

The present study showed the feasibility of using Auger emitters as theranostic agents. Specifically, we were able to show that the radiolabeled PARP inhibitor KX1 is capable of delivering iodine-125 and iodine-123 Auger emitting radionuclides to the nucleus of cells to induce DNA damage. Reduced cytotoxicity, and γH2AX foci observed after treating OVCAR8 PARP1 knockout cell lines demonstrated that both Auger emitters caused DNA damage as a result of specifically targeting PARP-1. These data provide evidence that radiolabeled PARP inhibitors such as KX1 can deliver Auger emitting radionuclides within close proximity to DNA to maximize the high-LET effects of Auger electrons for inducing DNA damage and cell death, building on our previous work [19].

Another interesting result found in our study is that DNA damage induced by KX1 is enzymatically independent of PARP-1 inhibition. The primary difference between radiolabeled PARP inhibitors and conventional PARP inhibitors as anticancer drugs is that radiolabeled PARP inhibitors induce DNA damage using ionizing radiation and that they do not enzymatically inhibit PARP-1. This is in contrast to conventional PARP inhibitors that primarily work through synthetic lethality, where loss of primary homologous recombination (HR) genes such as *BRCA1* or *BRCA2* combined with PARP inhibition results in cell death. Although PARP inhibition and *BRCA* mutations may confer synthetic lethality, the loss of HR also reduced cellular fitness against DNA damage caused by Auger radiation. This effect enabled [^125^I]KX1 to induce DNA damage and cell death dependent on *BRCA1* mutation, as shown in dose–response studies where restoration of *BRCA1* in UWB1.289 ovarian cancer cells decreased radiosensitivity. Indeed, *BRCA1* mutant ovarian cancer cell lines were the most sensitive to [^125^I]KX1; however, dosimetric analysis revealed interesting differences in RBE, which suggests [^125^I]KX1 is more effective than low-LET beta-emitting analogue [^131^I]KX1 in ovarian cancer cell lines with functional *BRCA1*. We propose that increased DNA repair capacity of ovarian cancer cell lines provides a better cellular fitness capable of mitigating DNA damage induced by low-LET radiation, whereas a known property of high-LET radiation is that cell kill effects are independent of DNA repair capacity. Together these data further support Auger electrons as a form of high-LET radiation that are less susceptible to common resistance mechanisms to low-LET radiation, even clinical PARP inhibitors. We found the *BRCA1* mutation resulted in a resistance enhancement of only 1.59 for [^125^I]KX1, in contrast to [^131^I]KX1, which showed 3.79, and is in stark contrast with the 100 to 1000 times difference for clinically used PARP inhibitors. Furthermore, *BRCA1* reversion mutations are the most common cause of clinical resistance where *BRCA1* regains partial or complete functions and tumors no longer respond. Our work shows [^125^I]KX1 has the potential to remain effective even after a *BRCA1* reversion mutation and possibly have efficacy in non-*BRCA* mutant tumors.

In this study, we showed that it is feasible to evaluate the radiosensitivity of viable patient tumor samples ex-vivo. This not only increases the translational potential of evaluating Auger-emitting radiopharmaceuticals for therapy but also enables the direct evaluation of anti-tumor effects before early phase clinical trials take place. Future work will be directed towards ex-vivo analysis of response to better understand tumor radiobiology and rigorously test the appropriateness for Auger-emitting radiopharmaceutical therapy. In summary, we have shown that radiolabeled PARP inhibitors can effectively and specifically serve as targeting vectors for the delivery of therapeutic Auger emitting radionuclides as cancer theranostics. Future studies will focus on the theranostic application of [^123^I]KX1, evaluating the predictive capability of SPECT imaging-based dosimetry on anti-tumor response and long-term survival in pre-clinical models.

## 4. Materials and Methods

### 4.1. PARP-1 Expression in Breast and Ovarian Cancer

To determine the relative expression of PARP-1 in breast or ovarian cancer vs. adult normal tissue, we used RNA sequencing data from The Cancer Genome Atlas (https://cancergenome.nih.gov/) and the Genotype Tissue Expression (GTEx) project.

### 4.2. Cell Culture

Cell lines were cultured using standard techniques under 5% CO_2_ and 10% O_2_ at 37 °C. OVCAR8 and OVCAR8 PARP1 knockout (KO) cell lines were cultured in RPMI 1640 with 10% FBS and 1% penicillin/streptomycin. Cell lines with stable knockout of PARP1 were previously reported [16]. Cells were selected for stable expression of each sgRNA using 2µg/mL puromycin for one week. PARP1 deletion was confirmed by Western blot protein analysis. OVCAR8 PARP1 KO were continuously selected with 2 µg/mL of puromycin. UWB1.289 and UWB1.289 *BRCA1* restored cell lines were cultured in a 1:1 mixture of MEGM with bullet kit (ATCC) and RPMI 1640 with 10% FBS and 1% penicillin/streptomycin.

### 4.3. Radiochemistry

Iodine-123 and iodine-125 were purchased from Nuclear Diagnostic Products Radiopharmacy (Cherry Hill, NJ) and Perkin Elmer (Waltham, MA). [^123/125^I]KX1 radiolabeling was performed as previously described by electrophilic destannylation of a tin precursor under mild oxidative conditions, although newer improved approaches have been reported for copper-mediated halodeboronation of boronic pinacol ester precursors [15]. Briefly, 100 µg of stannous-KX1 precursor material was dissolved in 50 µL of MeOH followed by the addition of 100 µL of 3:1 acetic acid:hydrogen peroxide. The reaction was heated at 100 °C for 30 min and purified by reverse phase chromatography under isocratic conditions (40% MeCN:60 % 1 M ammonium formate pH 4.5) on a semi-preparative HPLC using a Phenomenex Luna^®^ 5 µm C18 100 Å column 250 × 4.6 mm (Waters, Milford, MA, USA). The product peak was collected, diluted with water to <10% MeCN, and concentrated on a Sep-Pak C18 Plus Light Cartridge, 130 mg, 55–105 µm (Waters, Milford, MA, USA). The final product was diluted in 200 proof ethanol and diluted to a final concentration of <1% ethanol for respective biological studies.

### 4.4. Western Blot Analysis

OVCAR8 cells were treated with 3.7 MBq/mL of [^123^I]KX1 for either 2 or 24 h to determine whether [^123^I]KX1 treatment resulted in biochemical inhibition of PARP-1. OVCAR8 cells were treated with the FDA approved PARP inhibitor, olaparib, as a positive control. Cell lysates were prepared by first lysing cells in RIPA buffer (Thermo Fisher, Philadelphia, PA, USA) with protease and phosphatase inhibitors (Sigma Aldrich, St. Louis MO) on ice for 30 min followed by sonication. Lysates were then centrifuged at 14,000× *g* for 20 min at 4 °C to remove cellular debris. Next, solutions were diluted with 4× Laemmli buffer and heated at 100 °C for 5 min. Gel electrophoresis was performed on a BioRad system and transferred to a PDVF membrane using a BioRad Turbo transfer at 1.3 A 25V for 7 min. Membranes were then blocked in LiCor Odyssey blocking buffer for 1 h, followed by 1 h incubation at 37 °C in Odyssey blocking buffer with primary antibodies for PARP-1 (1:1000, Cell Signaling and Technologies 46D11), and PAR/poly (ADP-ribose) (1:1000, Enzo 10H), γH2AX (1:5000, Millipore JBW301). Membranes were then washed 3 times for 5 min per wash in 0.2% PBST. Secondary antibodies corresponding to primary species (Thermo Fisher, Philadelphia, PA, USA) were added at a 1:10,000 dilution in Odyssey blocking buffer and incubated at 37 °C for 1 h. Membranes were then washed 3 times for 5 min per wash in PBST followed by 1 wash in PBS. Membranes were then read on a LiCor imager.

### 4.5. Single Cell Immunofluorescence

OVCAR8 cells were analyzed by immunofluorescent cell microscopy to measure PARP-1 inhibition and the isogenic pair, OVCAR8 and OVCAR8 PARP1 KO, cell lines to evaluate [^123/125^I]KX1 induced DNA damage. Cells were first seeded on round cover slips in 24-well plates at 50,000 cells/well and 8-well chamber slides at 10,000 cells/well for 48 h before treatment with [^123/125^I]KX1 at concentrations from 0.74–4.44 MBq/mL for 1, 2, or 24 h. After treatment, cells were washed once with PBS, then fixed with 2% PFA for 10 min, washed with PBS 3 times, and permeabilized in 0.4% triton X on ice for 10 min. Cells were incubated with primary antibody incubation for 1 h, followed by 3 washes with 0.2% PBST, then incubated with corresponding secondary antibodies (goat anti-mouse IgG Alexa Fluor Plus 555 cat # A32727, goat anti-rabbit IgG Alexa Fluor Plus 488 cat # A327731) for 1 h. Cells were mounted with ProLong Glass mounting media containing NucBlue (Invitrogen). Primary antibodies used for analysis included DNA damage marker γH2AX (dilution: 1:5000, 05-636, anti-phospho-Histone H2A.X ser139, Millipore). The biochemical function of PARP-1 was analyzed through measuring the biochemical product poly(ADP-ribose) (PAR) (dilution: 1:1000, poly(ADP-ribose) monoclonal antibody 10H, Enzo, Farmingdale, NY, USA) after treatment with [^123/125^I]KX1. PARP-1 (dilution: 1:1000, PARP rabbit mAb 46D11, Cell Signaling Technology, Danvars, MA, USA) was assessed to confirm the absence of PARP-1 in OVCAR8 PARP1 KO cells. Nuclear staining of DAPI was used for identifying the nuclei of cells for quantifying DNA damage response markers. Images were taken on a Zeiss Observer Microscope and fluorescent intensity was quantified using Zeiss Zen software (Zeiss, Netherlands). For confocal microscopy experiments, images were acquired on a Leica SP8 Confocal Microscope and fluorescence intensity was quantified using Cell Profiler software.

### 4.6. Cell Viability

OVCAR8, OVCAR8 PARP1 KO Guide 1(G1), OVCAR8 PARP1 KO G2, OVCAR8 PARP1 KO G3, UWB1.289, and UWB1.289-*BRCA1* restored cell lines were seeded at 1000 cells/well in a 96-well plate for 24 h and subsequently treated with [^125^I]KX1 at doses ranging from 37–3700 MBq/mL for 1 h. Following the treatment period, treatment medium was aspirated, and cell culture media was replenished. Cells were then allowed to regrow for 5–7 days and cell viability was quantified by measuring ATP using the bioluminescent assay CellTiter Glo (Promega, Waltham, MA, USA). Plates were read on a Perkin Elmer EnSpire multimode plate reader (Waltham, MA, USA). Dose–response curves were fitted using a non-linear sigmoidal dose–response curve in Prism GraphPad v 7.0. Effective concentrations for 50% reductions in cell viability were calculated.

### 4.7. Radiation Dosimetry

To calculate the radiation dosimetry of [^131^I]KX1 and [^125^I]KX1, methods were performed as previously published [19]. Briefly, the radiation dose to the cell nucleus was derived from the radiopharmacology data (Appendix A) and calculated using Monte Carlo simulation with Medical Internal Radiation Dosimetry Cell (MIRDcell) V2.1, as previously described [20]. The radii of the cell and its nucleus were measured using phase contrast and fluorescent microscopy with DAPI. The cytotoxic dose–response curves for the radiopharmaceuticals were transformed to radiation dose–response curves based on the linear-quadratic model; 50% survival was used as the reference endpoint. The relative biological effectiveness (RBE) of each type of radiation was calculated, and data are reported as [^125^I]KX1 RBE/[^131^I]KX1 RBE.

### 4.8. SPECT/CT Imaging, Ex-Vivo Autoradiography and Histology

All animal studies were conducted under protocols approved by the University of Pennsylvania Institutional Animal Care and Use Committee. Tumor xenografts were generated by injecting 1 × 10^7^ OVCAR8 cells subcutaneously into the right flank of 10 week-old female SCID mice that weighed between 20–25 grams. Tumors were allowed to reach 100 mm^3^ over 4 weeks before SPECT/CT imaging studies were performed. On the day of the study, [^123^I]KX1 was injected at 29.6 MBq/mouse (*n* = 3). SPECT/CT imaging was performed on the U-SPECT and U-CT (Netherlands) from 40–120 min post injection. Images were co-registered using PMOD version 3.7 and tumor to muscle ratios were calculated. Immediately following the completion of imaging, tumors and muscle controls were resected, snap frozen, and sectioned at 30 microns on a Leica cryostat (Wetzlar, Germany). Sections were then exposed to phosphor films for 24 h and then the film was read on a Perkin Elmer Typhoon (Waltham, MA, USA). Tumor to muscle ratios were then calculated using Perkin Elmer software. Adjacent portions of the tumors were immediately fixed post resection for histological analysis.

### 4.9. Viable Tumor Slice Cultures

Tumors were obtained from patients at the time of ovarian cancer biopsy (IRB #702679). Fresh viable tumor samples were obtained from the PENN Ovarian Cancer Research Center Biotrust Collection (https://www.med.upenn.edu/OCRCBioTrust/). Tissues were embedded in low melting point agarose and viably sectioned at 300 µm using a VF-310-0Z vibrating microtome (Precisionary Instruments, Greenville, NC, USA). Tumor slices were cultured in complete media (DMEM/F12 + 10% FBS + 1% penicillin/streptomycin) overnight. Media was replaced and cells were treated with 25, 50, or 3.7 MBq/mL [^125^I]KX1 with or without the addition of 500 nM veliparib. Tissue slices were fixed and stained with primary and secondary antibodies as described above and mounted on glass slides for confocal microscopy.

### 4.10. Histology

Murine tumors were fixed in 10% neutral buffered formalin for 48 h followed by washing with PBS and dehydrating in a 20% glucose solution for 24 h. After dehydration, tumors were washed with PBS, embedded in OCT, and were sectioned on a cryostat at a thickness of 5 µm. Adjacent tissue sections were stained for hematoxylin and eosin and RPA (dilution 1:1000, NA19L anti-replication protein A (Ab-3) mouse mAb RPA34-20, Millipore) with DAPI mounting medium. Tissue sections were imaged on a Zeiss Observer microscope with motorized stage at 10×. Images were then reconstructed into whole section images using Zeiss Zen software.

Patient tumor slices were fixed in 10% neutral buffered formalin for 48 h followed by washing with PBS then embedded in paraffin wax. Tissues were dehydrated in graded ethanol solutions, cleared in xylene, then embedded in paraffin. Blocks were cut into 5 μm sections and stained using the DAKO CoverStainer for H & E (Agilent, Santa Clara, CA, USA). Immunohistochemistry was performed using the Leica Bond-IIITM with the Bond Polymer Refine Detection System. Then, the tissue was dehydrated, and antigen retrieval was optimized using sodium citrate, pH 6.0 or EDTA, pH 9.0. Primary antibodies used were cleaved-PARP and PARP (Cell Signaling, 46D11). Images were acquired using a Leica DM 2000 microscope. For quantification of tumor-specific cleaved-PARP+, the average number of pixels was calculated from 3 to 4 40× images using Aperio ImageScope software.

### 4.11. Statistics

When appropriate, results were reported as mean ± standard deviation (SD) or mean ± standard error (SE) unless denoted otherwise. Data were analyzed using one-way ANOVA with Dunnett’s multiple comparisons post-hoc test, or two-way ANOVA with Tukey’s multiple comparisons post-hoc test, or unpaired Student’s *T*-test. Statistical significance was set at * < 0.05, ** < 0.01, *** < 0.001, **** < 0.0001 (GraphPad Prism, La Jolla, CA, USA). In-vitro experiments were repeated at least three times with two or more replicates per experiment.

### 4.12. Radioligand Binding Assays

OVCAR8 wt and PARP1 KO cells were plated at 50,000 cells/well in 96-well plates 24 h before the assay in complete growth medium. For saturation curves, cells were incubated with increasing concentrations of [^125^I]KX1 (0.2–50 nM), and allowed to incubate until equilibrium was reached at 1 h. Then, 10 μM veliparib was used to determine non-specific binding. For competitive inhibition curves, 0–500 nM veliparib was co-incubated with [^125^I]KX1 at its K_d_, and allowed to incubate until equilibrium was reached at 1 h. Following incubation, media was aspirated, cells were washed with PBS, and radioactivity was measured on a Perkin Elmer Wizard gamma counter. Competition and saturation curves were plotted using GraphPad Prism version 6.0 and competitive inhibition constants (K_i_) and saturation dissociation constants (K_d_) were calculated.

## Figures and Tables

**Figure 1 molecules-25-06029-f001:**
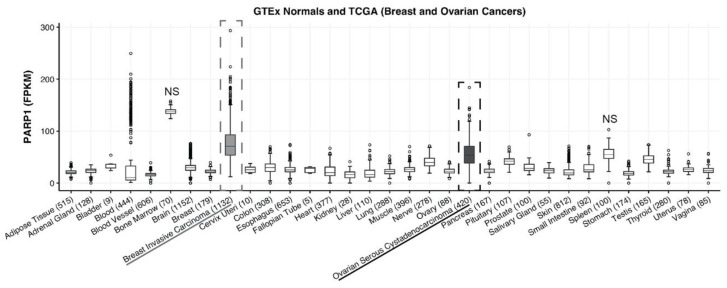
Poly (ADP-ribose) polymerase 1 (PARP-1) expression is increased in ovarian and breast cancer. RNA sequencing results were acquired through the publicly available Cancer Genome Atlas. Patient sample size is listed in parentheses next to each tissue type; 1132 breast (grey dotted box) and 420 ovarian cancer (black dotted box) patients’ tumors vs. adult normal tissue showed a statistically significant higher expression of PARP-1 in both cancers compared to all normal tissues except bone marrow and spleen (one-way ANOVA; *p*-value < 0.05).

**Figure 2 molecules-25-06029-f002:**
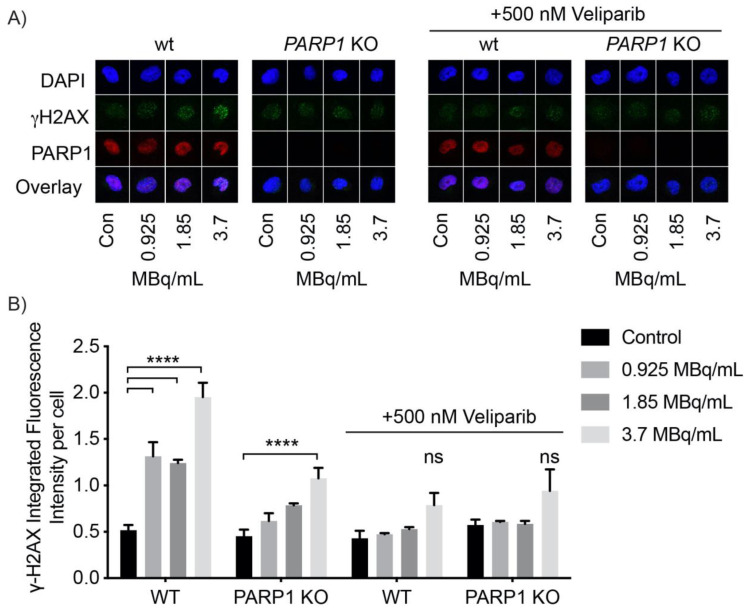
Immunofluorescent analysis of PARP-1 specific DNA damage induced by [^125^I]KX1 in OVCAR8 wt and PARP1 KO cells. (**A**) Images of representative single cells treated with [^125^I]KX1 with and without veliparib blocking. (**B**) Quantification of DNA damage marker γH2AX (one-way ANOVA with Dunnett’s multiple comparisons post-hoc test; *p*-value < ****0.0001).

**Figure 3 molecules-25-06029-f003:**
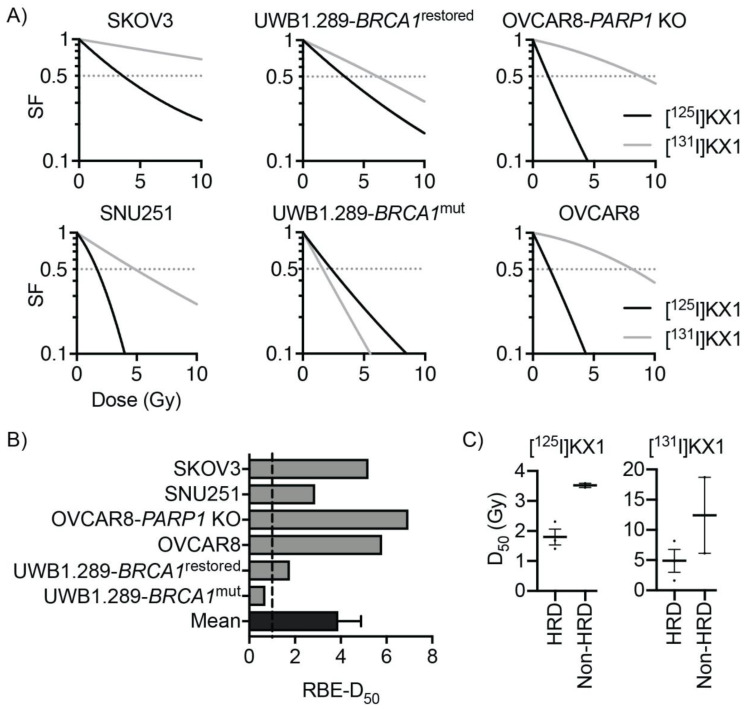
In-vitro radiation dose–response and relative biological effectiveness of [^125^/^131^I]KX1 in ovarian cancer cell lines. (**A**) Cell survival curves showing sensitivity to radiation dose deposited by PARP-1 targeted Auger emitter [^125^I]KX1 and beta emitter [^131^I]KX1. Horizontal dashed lines indicate 50% cell survival. (**B**) Relative biological effectiveness (RBE) of [^125^I]KX1 compared to [^131^I]KX1 based on 50% cell survival. RBE of [^125^I]KX1 greater than 1 was observed in 5 out of 6 cell lines, resulting in mean ± SEM of 3.9 ± 1.0. *p* < 0.05 from two-tailed one sample t-test. (**C**) Grouped comparison of HRD and non-HRD ovarian cancer cell lines for radiosensitivity to [^125^I]KX1 and [^131^I]KX1. HRD cell lines were more sensitive than non-HRD cell lines to [^125^I]KX1 (unpaired Student’s *t*-test; *p*-value < 0.05).

**Figure 4 molecules-25-06029-f004:**
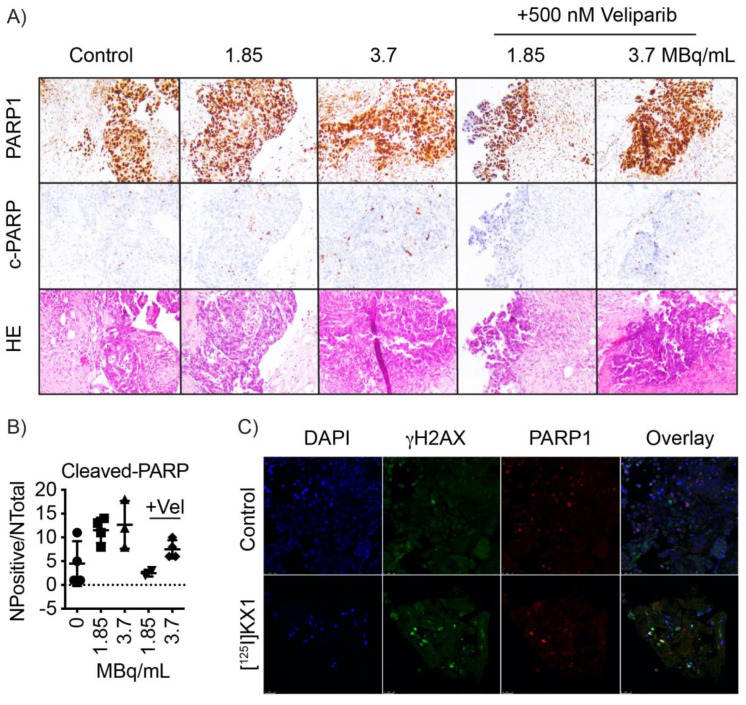
[^123^I]KX1 induces apoptosis in viable human tumor slices. Tumors from donors with high-grade serous ovarian cancer were viably sectioned using a compresstome. Tumor slices were cultured with 1.85 or 3.7 MBq/mL of [^125^I]KX1 with or without 500 nM veliparib for 2 h. (**A**) Tumors were formalin fixed and processed for immunohistochemistry. Serial sections were stained for PARP, cleaved-PARP or HE. (**B**) Quantification of positive cleaved-PARP staining pixels normalized to the area of tumor. (**C**) Immunofluorescence of tissue slices treated with 3.7 MBq/mL of [^125^I]KX1 for 1 h and assessed for DNA damage.

**Figure 5 molecules-25-06029-f005:**
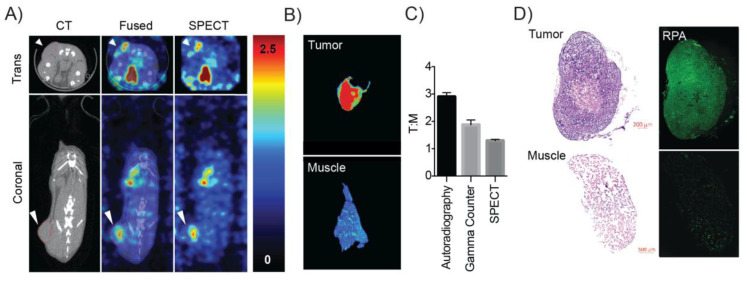
SPECT/CT imaging of [^123^I]KX1 with ex-vivo corresponding autoradiography and tissue histology. (**A**) SPECT/CT image of an OVCAR8 tumor bearing mouse showed elevated uptake in the tumor compared to surrounding tissue, the white arrow corresponds to the tumor. (**B**) Ex-vivo autoradiography of tumor vs. muscle showed greater uptake in the tumor. (**C**) Using three separate measures of tumor to muscle ratios we found differences in instrument sensitivity corresponded to better delineation of tumor uptake compared to normal tissues with the highest tumor to muscle uptake ratio obtained from autoradiography. (**D**) Ex-vivo histology of tumors after imaging showed tumors were positive for DNA damage marker RPA and muscles showed less. There was a tumor-dependent uptake of the radiotracer that corresponded to higher DNA damage. Untreated control tumors showed reduced RPA staining compared to treated tumors. These data provide the proof of concept for this approach and translate our in-vitro findings in-vivo.

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
