# Peer review of "PARP Theranostic Auger Emitters Are Cytotoxic in BRCA Mutant Ovarian Cancer and Viable Tumors from Ovarian Cancer Patients Enable Ex-Vivo Screening of Tumor Response"

_molecules, 2020, doi:10.3390/molecules25246029_

Round 1

Reviewer 1 Report

I think that submitted manuscript includes interesting findings, however, there are a few minor points which I think should be addressed:

Line 94: Add the citation

Line 102: Change ‘Supplmentary’ to ‘Supplementary’

Line 103: Add the citations

Lines 111-122: The saturation and competition radioligand binding assays should be added in the "Materials and Methods" section

Line 149: Change ‘10uM’ to ‘10 μM’

Line 162: Change ‘activitywas’ to ‘activity was’

Subsection 2.6: Why is the ATP-based luminiscent viability assay used as method to evaluate cell survival curves when "Clonogenic assay" has been described as the gold standard for radiation-induce cell death?. Which is the advantage of the used method?

Line 216: Change ‘obtined’ to ‘obtained’

Line 364: Change ‘hou.’ to ‘hour.’

Line 375: Delete ‘(cite:’

Line 388: Recommend adding “PMOD Version X.X”

Line 398: Change ‘melting point auger’ to ‘melting point paraffin (or agarose, etc)’

Sup Figure 1: Change ‘nM Radiolgiand’ to ‘nM Radioligand’ in B)

Sup Figure 2 and 3: There are inconsistencies on how activity units for a radioactive source are written in the manuscript. In some instances they are written as Bq and in these Figures as μCi. Becquerel (Bq) should be used.

Sup Table 1: Data should be expressed as Mean ± SD

Sup Table 3. Data should be expressed as Mean ± SD (no SEM), as it is detailed in Subsection 4.11 Statistics

Author Response

Line 94: Add the citation

Authors response - citations added

Line 102: Change ‘Supplmentary’ to ‘Supplementary’ 

Authors response - corrected

Line 103: Add the citations

Authors response - citations added

Lines 111-122: The saturation and competition radioligand binding assays should be added in the "Materials and Methods" section 

Authors response - methods added at section 4.12

Line 149: Change ‘10uM’ to ‘10 μM’

Authors response - corrected

Line 162: Change ‘activitywas’ to ‘activity was’

Authors response - corrected

Subsection 2.6: Why is the ATP-based luminiscent viability assay used as method to evaluate cell survival curves when "Clonogenic assay" has been described as the gold standard for radiation-induce cell death?. Which is the advantage of the used method?

Authors response - The clonogenic assay has been historically accepted as the "gold standard" for measuring cytotoxicity to cells, however ATP-based luminescent assays are an appropriate assay for measuring cellular toxicity to drugs that induce DNA damage mediated cell death and work through non-metabolism related pathways which could interfere with ATP production and luminescent signal. In addition it provides the advantage of being high-throughput and able to characterize numerous concentrations with a quantitative read-out in units of luminescence. There is a linear relationship with cell viability, cell number, and luminescence. 

Line 216: Change ‘obtined’ to ‘obtained’

Authors response - corrected

Line 364: Change ‘hou.’ to ‘hour.’

Authors response - corrected

Line 375: Delete ‘(cite:’

Authors response - corrected

Line 388: Recommend adding “PMOD Version X.X”

Authors response - change to include "PMOD version 3.7"

Line 398: Change ‘melting point auger’ to ‘melting point paraffin (or agarose, etc)’

Authors response - corrected to "melting point agarose"

Sup Figure 1: Change ‘nM Radiolgiand’ to ‘nM Radioligand’ in B)

Authors response - Corrected

Sup Figure 2 and 3: There are inconsistencies on how activity units for a radioactive source are written in the manuscript. In some instances they are written as Bq and in these Figures as μCi. Becquerel (Bq) should be used.

Authors response - Corrected throughout all figures and supplemental figures

Sup Table 1: Data should be expressed as Mean ± SD

Authors response - Data has been now expressed as ± SD

Sup Table 3. Data should be expressed as Mean ± SD (no SEM), as it is detailed in Subsection 4.11 Statistics

Authors response - Data was correctly presented. Corrections have been made to the statistics section to include "When appropriate, results were reported as mean ± standard deviation (SD) or mean ± standard error (SE) unless denoted otherwise."

Reviewer 2 Report

The authors wrote a manuscript concerning the inverstigation of a new thernostic radiopharmacon based on iodine-123 and iodine-125 against PARP. The manuscript is well written and the experiments are explained clearly. Some minor remarks are found. Please revise it before publication.

First sentence of abstract is equal to the first sentence of introduction, could you change that.

In the whole manuscript: please check if "in vivo", "in vitro", and "ex vivo" is written in italics (or not) (please check instruction for authors).

Please check if the reference in brackets is located "inside" the sentence or after the "." (please check instructions for authors, it should be inside the sentence) see eg lines 61/267

References 14, 18, and 19 have not the complete reference information please complete it.

Would be nice to have citation for KX1 and [123/125I]KX1, line 94: citation is missing.

Could you please add a chemical srtrucutre of KX1 and the stannyl precursor, maybe there is a citation for the synthesis and radiolabeling? Please add it or alternatively use SI for these information.

Line 103: citation is missing???

Lines 112/113/119 and later: please check Kd, Ki, Bmax is written with subscript of "d", "i", and "max" (whole document)

Line 320: please delete "Iodine-"

Section 4.3. should briefly contain the radiolabeling method. With this note, the labeling procedure is not clear for me.

Line 364: either "hour" or "h" instead of "hou", please correct it

Line 375: bracket is missing ")" after [21]

Line 401: please at a space character "500 nM"

Please delete line 501 if not ...

Carefully check aims and scope of the journal, I would recommend to use another journal like "Cancers" or "Pharmaceuticals" also coming from mdpi.

SI: line 3: delete "Title"

Author Response

The authors wrote a manuscript concerning the inverstigation of a new thernostic radiopharmacon based on iodine-123 and iodine-125 against PARP. The manuscript is well written and the experiments are explained clearly. Some minor remarks are found. Please revise it before publication.

First sentence of abstract is equal to the first sentence of introduction, could you change that.

Authors Response - The first sentence of the abstract has been changed to "Theranostics are emerging as a pillar of cancer therapy that enable the use of single molecule constructs for diagnostic and therapeutic application."

In the whole manuscript: please check if "in vivo", "in vitro", and "ex vivo" is written in italics (or not) (please check instruction for authors).

Authors Response - Corrected throughout the manuscript

Please check if the reference in brackets is located "inside" the sentence or after the "." (please check instructions for authors, it should be inside the sentence) see eg lines 61/267

Authors Response - Corrected throughout the manuscript

References 14, 18, and 19 have not the complete reference information please complete it.

Authors Response - References have been corrected

Would be nice to have citation for KX1 and [123/125I]KX1, line 94: citation is missing.

Authors Response - Citation added

Could you please add a chemical srtrucutre of KX1 and the stannyl precursor, maybe there is a citation for the synthesis and radiolabeling? Please add it or alternatively use SI for these information.

Authors Response - Citation has been included for the radiosynthesis of I-125-KX1

Line 103: citation is missing???

Authors Response - Citation added

Lines 112/113/119 and later: please check Kd, Ki, Bmax is written with subscript of "d", "i", and "max" (whole document)

Authors Response - Corrected throughout the manuscript

Line 320: please delete "Iodine-"

Authors Response - Corrected 

Section 4.3. should briefly contain the radiolabeling method. With this note, the labeling procedure is not clear for me.

Authors Response - The radiochemistry methods section has been expanded as follows "[123/125I]KX1 radiolabeling was performed as previously described by electrophilic destannylation of a tin precursor under mild oxidative conditions, although newer improved approaches have been reported for copper-mediated halodeboronation of boronic pinacol ester precursors [15]. Briefly, 100 µg of stannous-KX1 precursor material was dissolved in 50 µL of MeOH followed by the addition of 100 µL of 3:1 acetic acid:hydrogen peroxide. The reaction was heated at 100 ˚C for 30 minutes and purified by reverse phase chromatography under isocratic conditions (40% MeCN:60 % 1 M ammonium formate pH 4.5) on a semi-preparative HPLC using a Phenomenex Luna® 5 µm C18 100 Å column 250 x 4.6 mm (Waters, Milford MA). The product peak was collected, diluted with water to < 10% MeCN, and concentrated on a Sep-Pak C18 Plus Light Cartidge, 130 mg, 55-105 µm (Waters, Milford MA) The final product was diluted in 200 proof ethanol and diluted to a final concentration of < 1% ethanol for respective biological studies."

Line 364: either "hour" or "h" instead of "hou", please correct it

Authors Response - Corrected throughout the manuscript.

Line 375: bracket is missing ")" after [21]

Authors Response - Corrected 

Line 401: please at a space character "500 nM"

Authors Response - Corrected 

Please delete line 501 if not ...

Authors Response - Corrected 

Carefully check aims and scope of the journal, I would recommend to use another journal like "Cancers" or "Pharmaceuticals" also coming from mdpi.

Authors Response - Thank you for the recommendation however this was an invited manuscript for a special edition "Recent Advances in Techniques with Radionuclide for Theranostic Drugs" and the scope states "This Special Issue will focus on the synthesis and/or evaluation of radiolabeled compounds containing dosimetry for their clinical application, radionuclide production and separation, and combination of a radionuclide with other modalities for not only radiotheranostics but also imaging as companion diagnosis". Our work is focused on the biological evaluation of radiolabeled compounds and is well aligned with the issues scope.

SI: line 3: delete "Title"

Authors Response - Corrected